# Impact of Inorganic Solutes' Release in Groundwater during Oil Shale In Situ Exploitation

**Qingyu Li** [1,2], **Laijun Lu** [1], **Quansheng Zhao** [2] **and Shuya Hu** [2,*]

1    College of Earth Sciences, Jilin University, Changchun 130021, China
2    College of Environmental Science and Engineering, Qingdao University, Qingdao 266071, China
*    Correspondence: 90shuya@qdu.edu.cn

**Abstract:** Oil shale can produce oil and shale gas by heating the oil shale at 300–500 °C. The high temperature and the release of organic matter can change the physical and mechanical properties of rocks and make the originally tight impervious layer become a permeable layer under in situ exploitation conditions. To realize the potential impact of the in situ exploitation of oil shale on groundwater environments, a series of water–rock interaction experiments under different temperatures was conducted. The results show that, with the increase of the reaction temperature, the anions and cations in the aqueous solution of oil shale, oil shale–ash, and the surrounding rock show different trends, and the release of anions and cations in the oil shale–ash solution is most affected by the ambient temperature. The hydrochemical type of oil shale–ash solution is $HCO_3$-$SO_4$-Na-K at 80 °C and 100 °C, which changes the water quality. The main reasons are that (1) the high temperature ($\geq$80 °C) can promote the dissolution of FeS in oil shale and (2) the porosity of oil shale increases after pyrolysis, making it easier to react with water. This paper is an important supplement to the research on the impact of the in situ exploitation of oil shale on the groundwater environment. Therefore, the impacts of in situ mining on groundwater inorganic minerals should be taken into consideration when evaluating in situ exploitation projects of oil shale.

**Keywords:** oil shale in situ exploitation; groundwater; water–rock interaction; inorganic minerals; hydrogeochemistry

## 1. Introduction

Due to the limited oil reserves on Earth, oil shale as an important alternative energy has become one of our research focuses. Significant reserves of oil shale exist globally, amounting to nearly four-times more than the world's proven conventional oil reserves [1–6]. Oil shale is mainly produced by surface retorting technology or in situ exploitation technology presently [7–9]. Compared with traditional surface retorting technology, in situ exploitation is a relatively environmentally friendly way to extract shale oil. It does not require mining, transportation, or ore processing. Instead, heat is directly supplied by thermal conduction or thermal radiation to the underground oil shale layer to generate shale oil and gas, and then, the pyrolyzed oil and gas are recovered [10–14]. Scientists around the world have put forward a variety of methods for the in situ exploitation of oil shale [15–18], but some effects of in situ oil shale exploitation on the aquifer layer have not been thoroughly studied.

Figure 1 shows the conceptual model for oil shale in situ exploitation. Under natural conditions, the oil shale layer and upper surrounding rock usually form a dense waterproof layer. However, fracturing and pyrolysis change the original stress of the overlying rock mass. When the stress exceeds the shear strength of the rock, the rock layer fractures, resulting in a series of water-conducting fracture zones. The in situ pyrolysis process causes the oil shale layer to gradually change from the previously water-tight layer or weakly permeable layer to a permeable layer [19–21]. This evolution may lead to a hydraulic connection between the oil shale mining layer and the surrounding aquifer, resulting in

different degrees of water–rock interaction between groundwater and oil shale, the ash and slag of oil shale after mining, and the surrounding oil shale rocks [22].

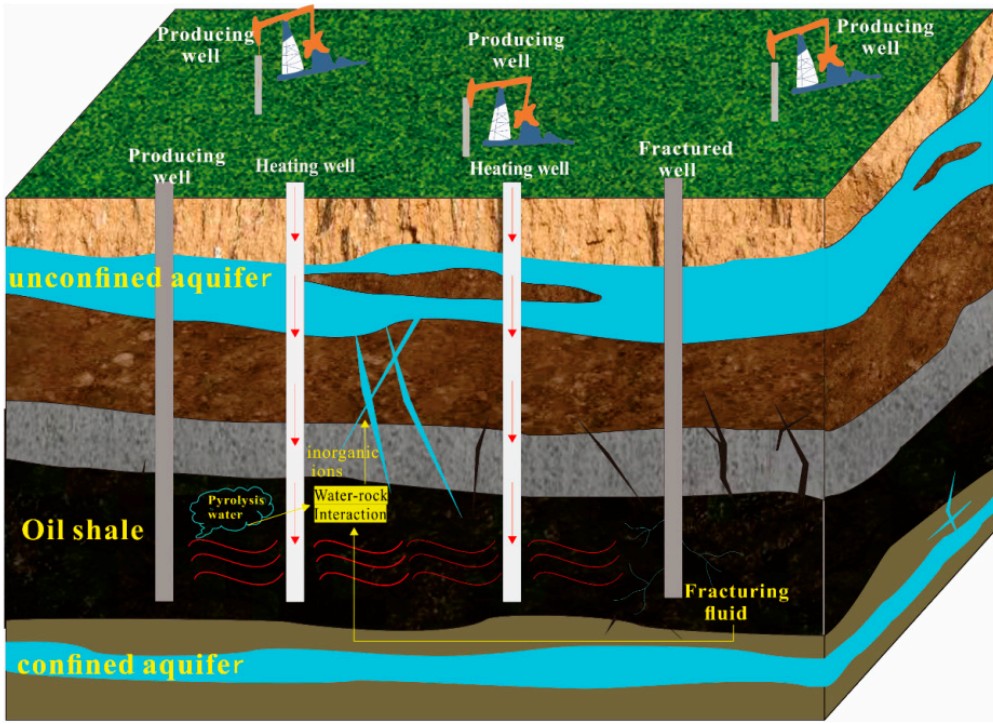

**Figure 1.** Conceptual model for oil shale in situ exploitation.

Regarding the effect of the in situ pyrolysis of oil shale on groundwater, a few studies have considered the potential pollution risk of water–rock interaction solutions in oil shale to groundwater, and a series of experimental studies have been carried out [19,22–25]. Hu [20] conducted a series of water–rock interaction experiments between oil shale–water and oil shale–ash–water to study the release of organic matter from groundwater during oil shale in situ exploitation. The results showed that the formation of fracture zones promoted by the pyrolysis of oil shale will cause organic contaminants to diffuse into deeper groundwater and affect the quality of groundwater. However, she did not discuss the effects of inorganic releases on groundwater. Wang [24] performed ultrapure water–rock interaction experiments to evaluate the release of heavy metals, i.e., Pb into the groundwater environment. He found that Pb tended to accumulate in solid residues during pyrolysis and then continued to be released in the groundwater during water–rock interactions. However, his experiments were conducted at room temperature and did not consider the effect of ambient temperature. However, the in situ exploitation of oil shale is a long process accompanied by heating. Studies have shown that oil shale exploitation affects the formation temperature field for a long time, and the in situ conversion process (ICP) utilizes electricity to heat the oil shale underground and needs to continue over 2 years [21].

Therefore, under long-term hydrothermal conditions, the release of inorganic minerals from oil shale to the groundwater environment should be studied thoroughly.

In this study, a series of water–rock interaction experiments including oil shale and water, oil shale–ash and water, and surrounding rock and water at different temperatures was designed to simulate the release of inorganic minerals during the in situ mining of oil shale. By identifying the mineral elements of the oil shale, oil shale–ash, or surrounding rock released into the groundwater, the changes in the water chemical types are discussed. In addition, the main reasons for the changes in water chemical types were explored through factor analysis. This work may fill the gap on the impact of the in situ exploitation of oil shale on the groundwater inorganic environment.

## 2. Materials and Methods

### 2.1. Study Area and Samples

The samples were collected from the Nong'an oil shale ore-bearing area, which belongs to the western edge of the northeast uplift belt of the Songliao Basin (Figure 2). This area is a stable geological environment with stable crust and relatively minimal tectonic activity. The oil yield of oil shale is generally 3.5–5% [19].

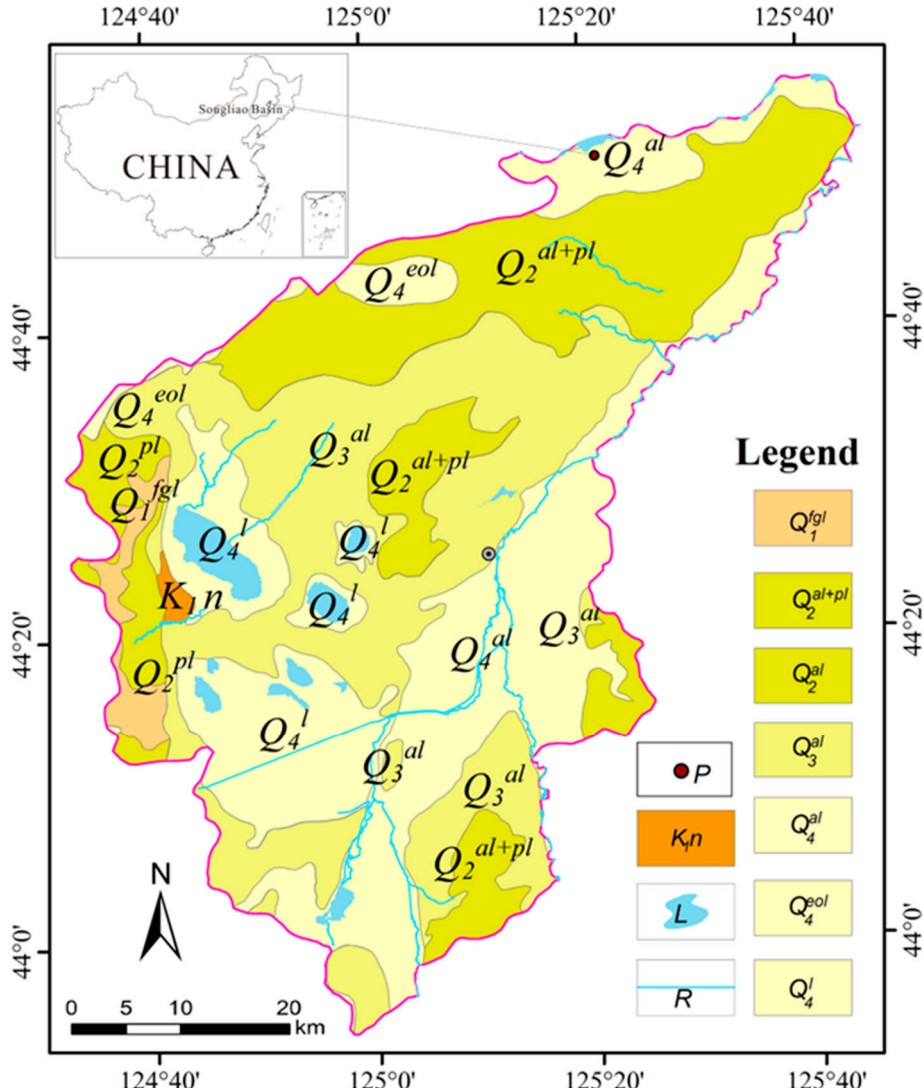

**Figure 2.** Geological map of Nong'an City [19]. ($Q_1^{fgl}$ represents the glaciofluvial accumulation layer of the Gelasian Quaternary Pleistocene; $Q_2^{al+pl}$ represents the alluvium and diluvium of the Middle Quaternary Pleistocene; $Q_2^{al}$ represents the alluvium of the Middle Quaternary Pleistocene; $Q_3^{al}$ represents the alluvium of the Upper Quaternary Pleistocene; $Q_4^{al}$ represents the alluvium of the Quaternary Holocene; $Q_4^{eol}$ represents the eolian deposit of the Quaternary Holocene; $Q_4^l$ represents the lake sediments of the Quaternary Holocene; $K_{1n}$ represents the mudstone or clastic rock of the Lower Cretaceous; L represents the lakes; R represents the rivers; P represents the sampling site of the oil shale in the area).

The stratified oil shale blocks and surrounding rock of the upper layer of oil shale were sealed and brought back to the lab, crushed, and screened to a size range of 2–3 cm to obtain experimental samples. The oil shale and surrounding rock were dried in an oven at 110 °C for 4 h before weighing. Some of the samples of oil shale and the surrounding

rock were further ground following XRD sample preparation requirements and subjected to XRD analysis to identify minerals and other crystalline phases in the samples.

### 2.2. Experiment of Water–Rock Interaction

The oil shale was heated to 400 °C in a tube furnace under nitrogen for 1 h to obtain oil shale–ash samples. Nitrogen was introduced to maintain an oxygen-free environment, since oil shale can only be pyrolyzed under anaerobic conditions. Ultrapure water (1 L) was added to 100 g of oil shale, oil shale–ash, and surrounding rock samples. The samples were placed in water baths with separation temperatures of $20 \pm 0.1$ °C, $50 \pm 0.1$ °C, $80 \pm 0.1$ °C, and $100 \pm 0.1$ °C. Each temperature tank had 10 samples, and they were independent of each other. Samples were soaked for 0.5, 1, 2, 5, 8, 15, 20, and 30 days. After soaking, the aqueous solution sample was filtered and tested.

### 2.3. Cation and Anion Content Determination

The calcium content in the solution was determined by ethylenediaminetetraacetic acid (EDTA) ($C_{10}H_{14}N_2O_8Na_2 \bullet 2H_2O$) titration. The contents of $K^+$, $Na^+$, and $Mg^{2+}$ were measured by a Shimadzu AA-6000CF (Shimadzu, Shanghai, China)atomic absorption spectrophotometer.

The content of carbonate and bicarbonate ions in the solution was measured by the acid standard solution titration method (national standard method DZ/T 0064.49-2021). The contents of $F^-$, $Cl^-$, $NO_3^-$, and $SO_4^{2-}$ in aqueous solution were determined by a Vantone 861 double-inhibitory ion chromatograph.

### 2.4. Analysis Method of Data

Factor analysis is a multivariate statistical analysis method that can be used to determine the hidden representative factors in many variables [26]. By using factor analysis, we can subsume a large number of variables with complex relationships into a few comprehensive factors to reveal the potential relationship between data [27,28].

Through the SPSS 25 software, general principal component extraction and variance maximum orthogonal rotation factor analysis were used to obtain the results presented in Table 1. The result of the KMO test was 0.553 (the KMO results > 0.5 indicate that the correlation of the variables meets the requirements for factor analysis). The result of the significance test was 0, indicating a correlation between the variables exists and the factor analysis could be carried out among the variables.

**Table 1.** KMO and Bartlett test result.

| Kaiser–Meyer–Olkin Test | | 0.553 |
|---|---|---|
| **Bartlett Test Result** | Approximate chi-squared | 1046.244 |
| | df | 78 |
| | Sig. | 0.000 |

## 3. Results

### 3.1. Mineral Composition of Samples

The XRD analysis of the oil shale from the Nong'an Formation revealed a complex mineral signature (Figure 3). The dominant mineral phases of the oil shale and the surrounding rock in Nong'an were roughly the same, and the main minerals were plagioclase ($Na[AlSi_3O_8]$-$Ca[Al_2Si_2O_8]$), illite ($KAl_2[(SiAl)_4O_{10}]\cdot(OH)_2\cdot nH_2O$), pyrite ($FeS_2$), calcite ($CaCO_3$), dolomite ($CaMg(CO_3)_2$), zeolite ($A_mB_pO_{2p}\cdot nH2O$), quartz ($SiO_2$), and montmorillonite ($(Na,Ca)_{0.33}(Al,Mg)_2[Si_4O_{10}](OH)_2\cdot nH_2O$). However, the contents of the different minerals in the oil shale and the surrounding rock were obviously different. For example, the content of pyrite ($FeS_2$) in the oil shale was higher than that in the surrounding rock, while the content of dolomite ($CaMg(CO_3)_2$) in the surrounding rock was higher than that in the oil shale. In addition, there were zeolite ($A_mB_pO_{2p}\cdot nH_2O$) and montmoril-

lonite $((Na,Ca)_{0.33}(Al,Mg)_2[Si_4O_{10}](OH)_2 \cdot nH_2O)$ in the surrounding rock, which were not detected in the oil shale samples.

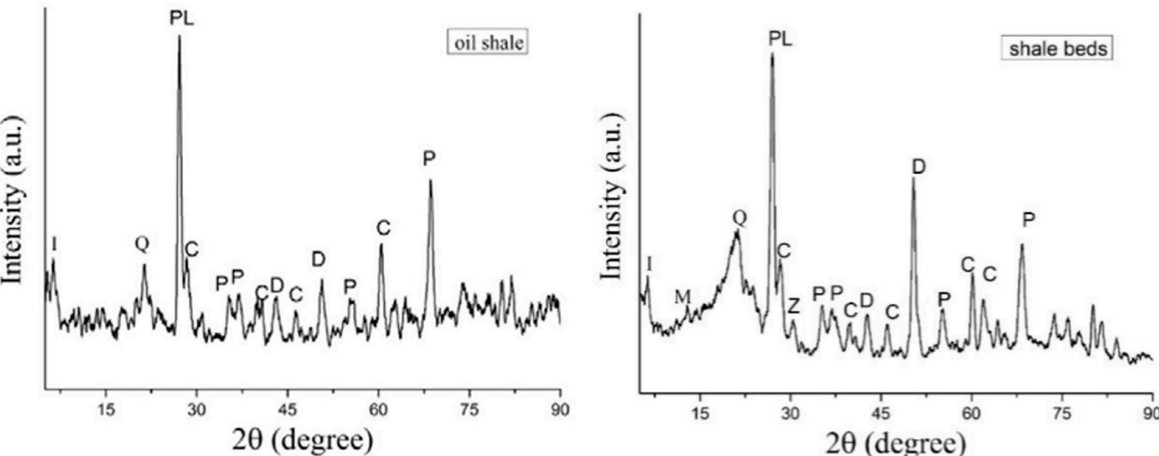

**Figure 3.** XRD spectra of the oil shale and shale beds. I, illite; Q, quartz; PL, plagioclase; C, calcite; P, pyrite; D, dolomite; M, montmorillonite; Z, zeolite.

### 3.2. Main Cation Variation Characteristics

Figure 4 shows the changes in the main cation content in the oil shale–water solutions at different reaction temperatures. There was little difference in the contents of the four cations in the aqueous solution under the reaction conditions of 20 °C and 50 °C. At reaction temperatures of 20 °C and 50 °C, the content of $Ca^{2+}$ was the highest, followed by $K^+$ and $Na^+$, and the content of $Mg^{2+}$ was the lowest. At reaction temperatures of 80 °C and 100 °C, the content of $Na^+$ in the oil shale–water solution changed strongly; the $Na^+$ content was the highest, followed by $K^+$ and $Ca^{2+}$, and the $Mg^{2+}$ content was the lowest, possibly because $Ca^{2+}$ and $Mg^{2+}$ in the aqueous solution came mainly from the dissolution of dolomite $(CaMg(CO_3)_2)$ and calcite $(CaCO_3)$. With increasing temperature, the solubility of dolomite $(CaMg(CO_3)_2)$ and calcite $(CaCO_3)$ decreased.

Moreover, $K^+$ changed little at 20 °C, 50 °C, and 80 °C, but its concentration increased rapidly at 100 °C, indicating that a temperature of approximately 100 °C had the greatest influence on the $K^+$ content. The cation content in the aqueous solution tended to be stable at approximately 15 days.

As shown in Figure 5, compared with the cation content in the oil shale aqueous solution, the cation content in the oil shale–ash aqueous solution showed a different trend with increasing reaction temperature. Under different reaction temperatures, the $K^+$ content in oil shale–ash aqueous solution always remained the highest, and the $K^+$ content at 50 °C, 80 °C, and 100 °C was twice the $K^+$ content at 20 °C. At 20 °C, the average content of $Na^+$ (5.79 mg/L) was lower than the average content of $Ca^{2+}$ (7.10 mg/L), but with increasing reaction temperature, the $Na^+$ content gradually increased and remained higher than the $Ca^{2+}$ content. $Ca^{2+}$ had a declining trend with increasing temperature. Under a reaction temperature of 100 °C, the $Na^+$ content in the oil shale–ash aqueous solution increased rapidly with increasing reaction time. After 30 days of reaction, the average contents of $Na^+$ and $K^+$ in the oil shale–ash aqueous solution were higher than those in the oil shale aqueous solution. This may be because high-temperature pyrolysis (>350 °C) precipitates organic matter in the oil shale, increases the pore space of the oil shale, and thus, increases the specific surface area of oil shale–ash for the reaction with the aqueous solution. On the other hand, high-temperature pyrolysis (>350 °C) will also break the structure of inorganic minerals and cause clay minerals to dewater, thus forming more pores [7].

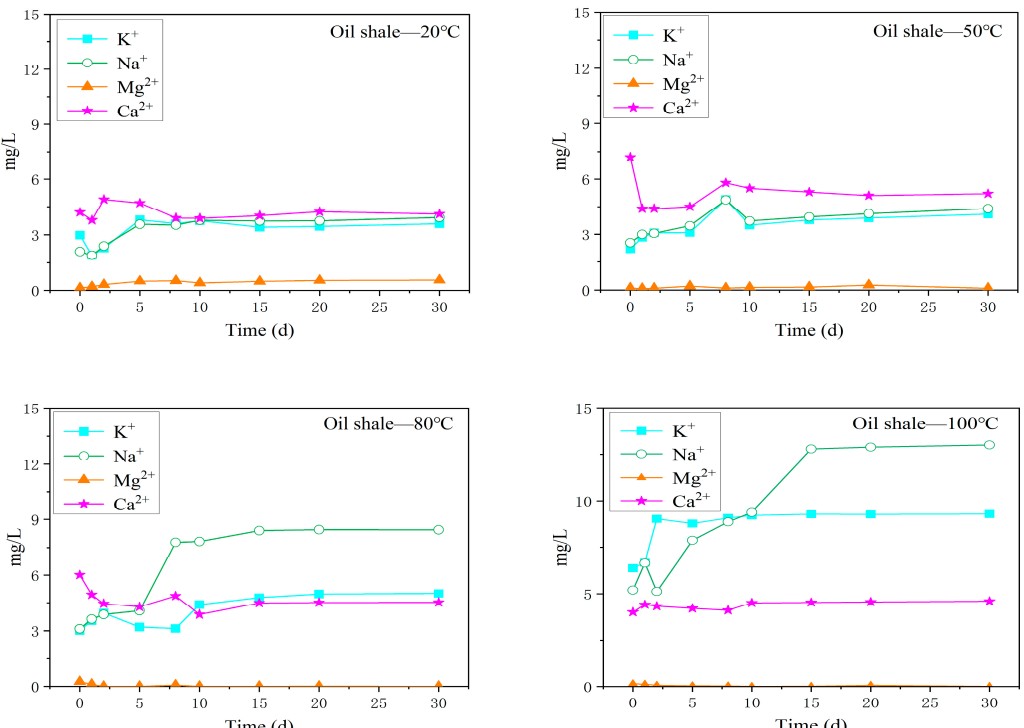

**Figure 4.** Variations in the content of the main cations in the oil shale aqueous solution under different temperature reaction conditions.

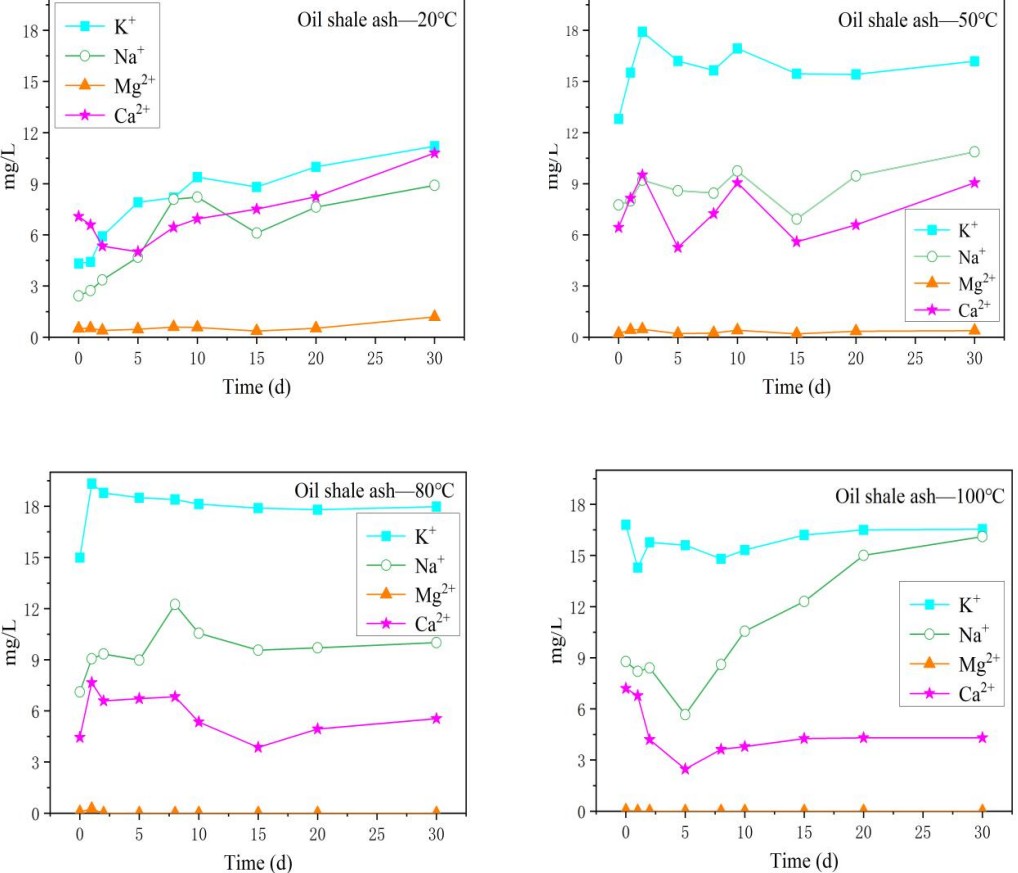

**Figure 5.** Variation in the content of the main cations in the oil shale–ash aqueous solution under different temperature reaction conditions.

The variation in the content of the main cations in the surrounding rock aqueous solution with different reaction temperatures is shown in Figure 6. With increasing reaction time, the content of cations showed an upwards trend. Under the reaction conditions of 20 °C and 50 °C, the content of $Ca^{2+}$ in the aqueous solution was the highest, followed by $Na^+$ and $K^+$, and $Mg^{2+}$ was the lowest. The content of $Na^+$ was the highest at 80 °C and 100 °C, followed by $Ca^{2+}$, $K^+$, and $Mg^{2+}$. With increasing reaction time, the cationic content in the aqueous solution increased steadily under the reaction condition of 80 °C, but fluctuated greatly under the reaction condition of 100 °C.

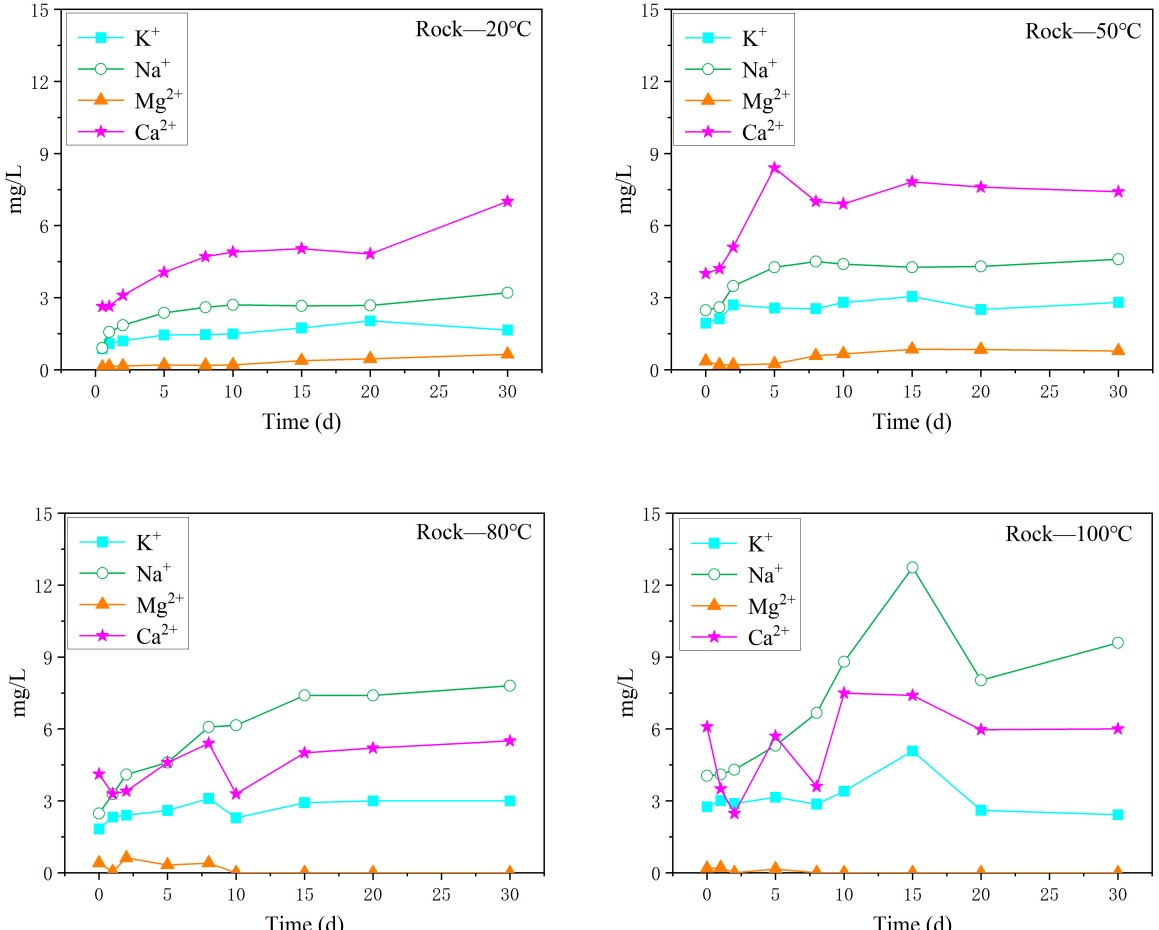

**Figure 6.** Variation in the content of the main cations in the oil shale–surrounding rock aqueous solution under different temperature reaction conditions.

### 3.3. Main Anion Variation Characteristics

The variation trend of the major anion content in the oil shale aqueous solution under different temperature reaction conditions is shown in Figure 7. The content of $HCO_3^-$ was always the highest, followed by $Cl^-$. When the reaction temperature was 20 °C, 50 °C, and 80 °C, the content of $Cl^-$ fluctuated, but increased significantly when the reaction temperature was 100 °C. After 30 days of reaction, the content of $Cl^-$ reached 15.4 mg/L. The content change of $SO_4^{2-}$ was similar to the content change of $Cl^-$, and the change was small at reaction temperatures of 20 °C, 50 °C, and 80 °C, but increased sharply at a reaction temperature of 100 °C. The contents of $F^-$ and $NO_3^-$ in the aqueous solution were always low at different temperature gradients.

The variation in the main anion content in the oil shale–ash–slag aqueous solution with reaction temperature is shown in Figure 8. $HCO_3^-$ was the main anion in the aqueous solution; the content of $HCO_3^-$ was greater than the content of oil shale aqueous solution, and the content was more than 20 mg/L. The content of $HCO_3^-$ decreased with increasing

reaction time in the aqueous solutions with higher reaction temperatures (80 °C and 100 °C). Different from the oil shale aqueous solution, the content of $SO_4^{2-}$ in the aqueous solution increased significantly, and the concentration of $SO_4^{2-}$ became the second-greatest anion, surpassing the content of $Cl^-$. With increasing reaction time, the content of $SO_4^{2-}$ also increased, especially after 30 days of reaction at 80 °C, and the content of $SO_4^{2-}$ in the aqueous solution reached 36.97 mg/L. $SO_4^{2-}$ in the aqueous solution generally came from the dissolution of minerals containing gypsum or other sulfates. In addition, the oil shale and its surrounding rock contained a large amount of pyrite. The oxidation of pyrite results in the presence of water-insoluble sulfur in water as $SO_4^{2-}$. At a 100 °C water temperature, the variation trend of $Cl^-$ was similar to that of $SO_4^{2-}$. The $F^-$ and $NO_3^-$ content was still low, and the $NO_3^-$ content was almost zero.

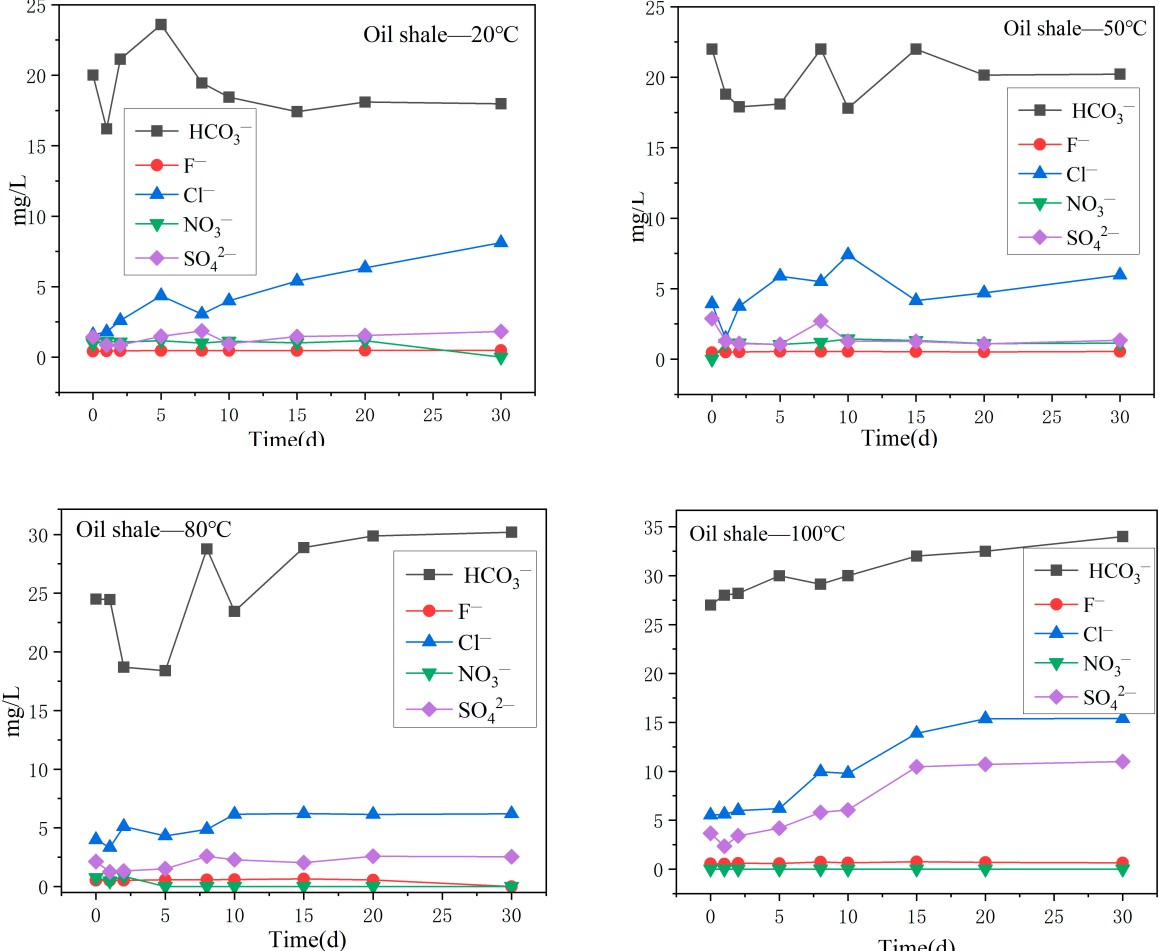

**Figure 7.** Variation in the content of the main anions in the oil shale aqueous solution under different temperature reaction conditions.

Figure 9 describes the changes in the main anion content in the surrounding rock aqueous solution at different reaction temperatures. Under the reaction conditions of 20 °C, 50 °C, and 80 °C, the variation trend of the main anions in the oil shale–surrounding rock water solution was similar to the variation trend of the main anions in the oil shale water solution. $HCO_3^-$ was the main anion, followed by $Cl^-$, and the content of $HCO_3^-$ and $Cl^-$ gradually increased with time. However, at the reaction temperature of 100 °C, the $HCO_3^-$ and $Cl^-$ content changed differently from other temperature conditions, and both of these ions increased first and, then, decreased with the reaction time. The content of $SO_4^{2-}$, $F^-$, and $NO_3^-$ in the aqueous solution did not change significantly and remained at a low level.

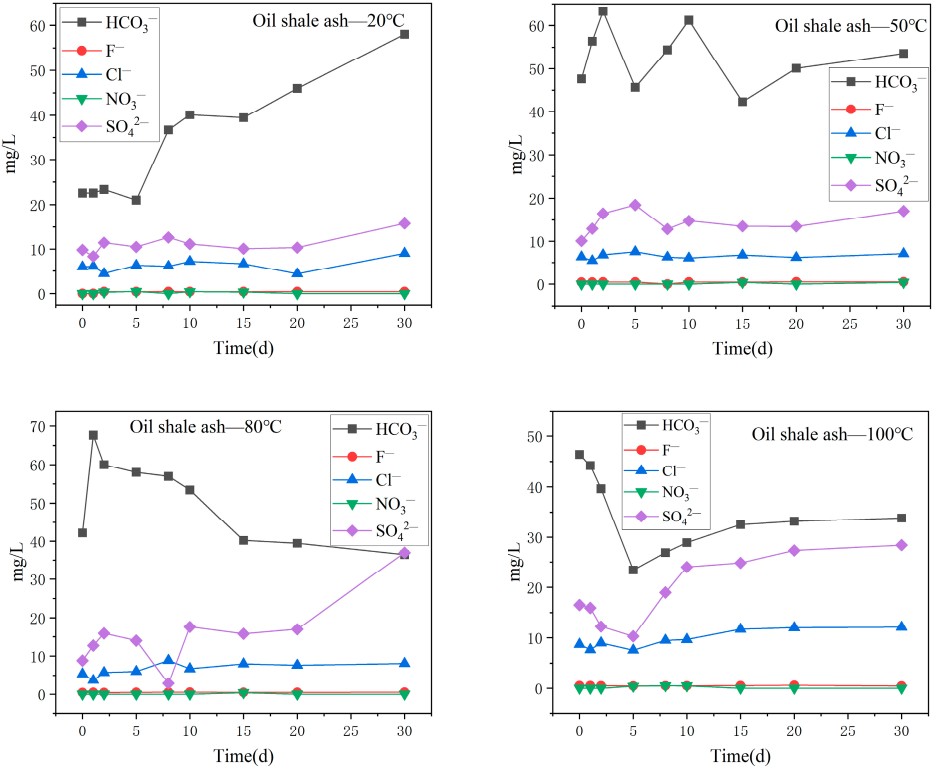

**Figure 8.** Variation in the content of the main anions in the oil shale ash aqueous solution under different temperature reaction conditions.

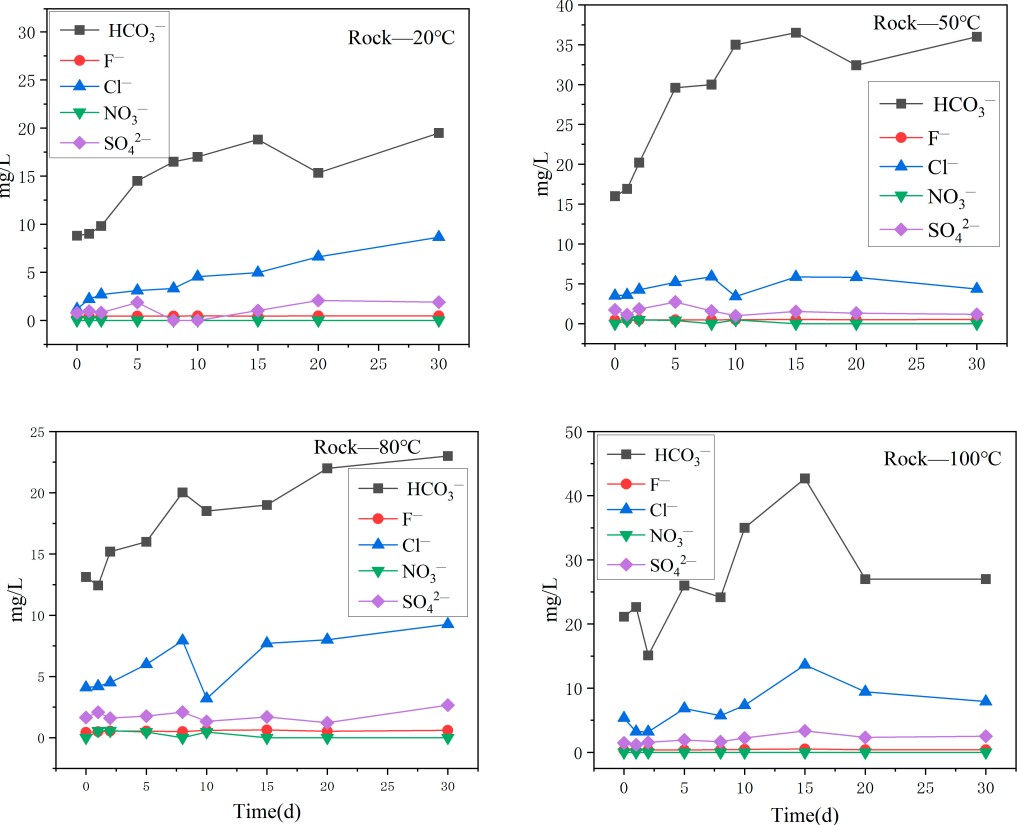

**Figure 9.** Variation in the content of the main anions in the surrounding rock aqueous solution under different temperature reaction conditions.

## 4. Discussion

### 4.1. Changes in the Water Chemistry

A Piper diagram can show the chemical composition of water–rock interactions in an aqueous solution, as shown by the Piper diagrams for different types of aqueous solutions (Figure 10). As shown in Figure 10a, in the oil shale aqueous solution, the dominant cations were $Ca^{2+}$ and $Na^+$, the dominant anions were $HCO_3^-$ and $Cl^-$, and the hydrochemical types were $HCO_3$-Cl-Ca-Na at 20 °C and 50 °C. At 80 °C, the dominant cations were $Na^+$ and $Ca^{2+}$, the dominant anion was $HCO_3^-$, and the hydrochemical type was $HCO_3$-Na-Ca. At 100 °C, the dominant cation was $Na^+$, the dominant anions were $HCO_3^-$ and $Cl^-$, and the hydrochemical type was $HCO_3$-Cl-Na. In the oil shale–ash aqueous solution (Figure 10b), the dominant cations were $Ca^{2+}$ and $Na^+$, the dominant anion was $HCO_3^-$ at 20 °C, and the hydrochemical type was $HCO_3$-Ca-Na. At 50 °C, the dominant cations were $Ca^{2+}$, $Na^+$ and $K^+$, the dominant anion was $HCO_3^-$, and the hydrochemical type was $HCO_3$-Ca-Na-K. At 80 °C and 100 °C, the dominant cations were $Na^+$ and $K^+$, the dominant anions were $HCO_3^-$ and $SO_4^{2-}$, and the hydrochemical type was $HCO_3$-$SO_4$-Na-K. In the surrounding rock aqueous solution (Figure 10c), the dominant cations were $Ca^{2+}$ and $Na^+$, the dominant anions were $HCO_3^-$ and $Cl^-$ at 20 °C, and the hydrochemical type was $HCO_3$-Cl-Ca-Na. At 50 °C, the dominant cations were $Ca^{2+}$ and $Na^+$, the dominant anion was $HCO_3^-$, and the hydrochemical type as $HCO_3$-Ca-Na. At 80 °C and 100 °C, the dominant cations were $Na^+$ and $Ca^{2+}$, the dominant anions were $HCO_3^-$ and $Cl^-$, and the hydrochemical type ws $HCO_3$-Cl-Na-Ca.

In Figure 10d, the grey arrows show that, (1) in the three aqueous solutions, the total contents of $Ca^{2+}$ and $Ca^{2+} + Mg^{2+}$ decreased gradually with increasing reaction temperature, indicating that calcite ($CaCO_3$) and dolomite ($CaMg(CO_3)_2$) precipitated from the aqueous solution with increasing temperature; (2) the $SO_4^{2-}$ contents of different aqueous solutions were significantly different. The content of $SO_4^{2-}$ in the oil shale–ash aqueous solution was the highest, followed by that in the oil shale aqueous solution, and the content of $SO_4^{2-}$ in the surrounding rock aqueous solution was the lowest. However, $SO_4^{2-}$ was mainly derived from the oxidation of pyrite ($Fe_2S$), indicating that a large amount of pyrite ($Fe_2S$) was released from the oil shale by the in situ pyrolysis of the oil shale and existed in the groundwater in the form of $SO_4^{2-}$ after interacting with the water, which would change the hydrochemical type of the groundwater from its original condition.

### 4.2. The Source of the Major Components

Table 2 shows the eigenvalues and cumulative variance contribution rates of the factor correlation matrix calculated by the factor analysis. It can be seen from the factor contribution rate that the eigenvalues of the first four factors were greater than 1, and the sum of the eigenvalues of the first four factors accounted for 75.223% of the total eigenvalues, which means 75.223% of the information of the total sample can be reflected by the four factors. Therefore, these four factors were extracted as the main factors.

**Table 2.** Total variance explanation.

| Common Factor | Initial Eigenvalue | | |
| | Eigenvalues | Variance Contribution Rate (%) | Cumulative Variance Contribution Rate (%) |
|---|---|---|---|
| 1 | 4.407 | 33.903 | 33.903 |
| 2 | 2.302 | 17.71 | 51.613 |
| 3 | 1.682 | 12.941 | 64.554 |
| 4 | 1.387 | 10.669 | 75.223 |

Table 3 shows the factor loading after rotation. The factor loading is the correlation coefficient between a variable and the factor. For a variable, the larger the absolute value of

the load is, the closer the relationship between it and the factor is. Therefore, the conclusions can be drawn from the factor loading as follows.

**Table 3.** Rotated factor loading matrix.

| Parameter | Factor 1 | Factor 2 | Factor 3 | Factor 4 | Commonality |
|---|---|---|---|---|---|
| $K^+$ | **0.939** | −0.125 | −0.016 | −0.054 | 0.901 |
| $Na^+$ | **0.784** | −0.318 | 0.456 | 0.088 | 0.931 |
| $HCO_3^-$ | **0.884** | 0.205 | 0.073 | 0.132 | 0.846 |
| $SO_4^{2-}$ | **0.876** | −0.078 | 0.023 | −0.052 | 0.777 |
| Temperature | 0.293 | **−0.784** | 0.187 | 0.223 | 0.786 |
| $Mg^{2+}$ | −0.106 | **0.881** | 0.047 | 0.043 | 0.791 |
| $Ca^{2+}$ | 0.488 | **0.66** | 0.142 | 0.215 | 0.74 |
| Time | 0.139 | 0.209 | **0.799** | −0.011 | 0.701 |
| $Cl^-$ | 0.506 | −0.279 | **0.658** | 0.091 | 0.775 |
| $F^-$ | −0.072 | −0.423 | **0.527** | −0.262 | 0.53 |
| $Fe^{2+}$ | −0.127 | −0.001 | **0.521** | 0.49 | 0.527 |
| Lithology | −0.141 | 0.123 | −0.094 | **0.824** | 0.723 |
| $NO_3^-$ | −0.354 | 0.139 | −0.063 | **−0.776** | 0.751 |

The numbers in bolds represent the variables have larger absolute values (closer relationships) with the factors.

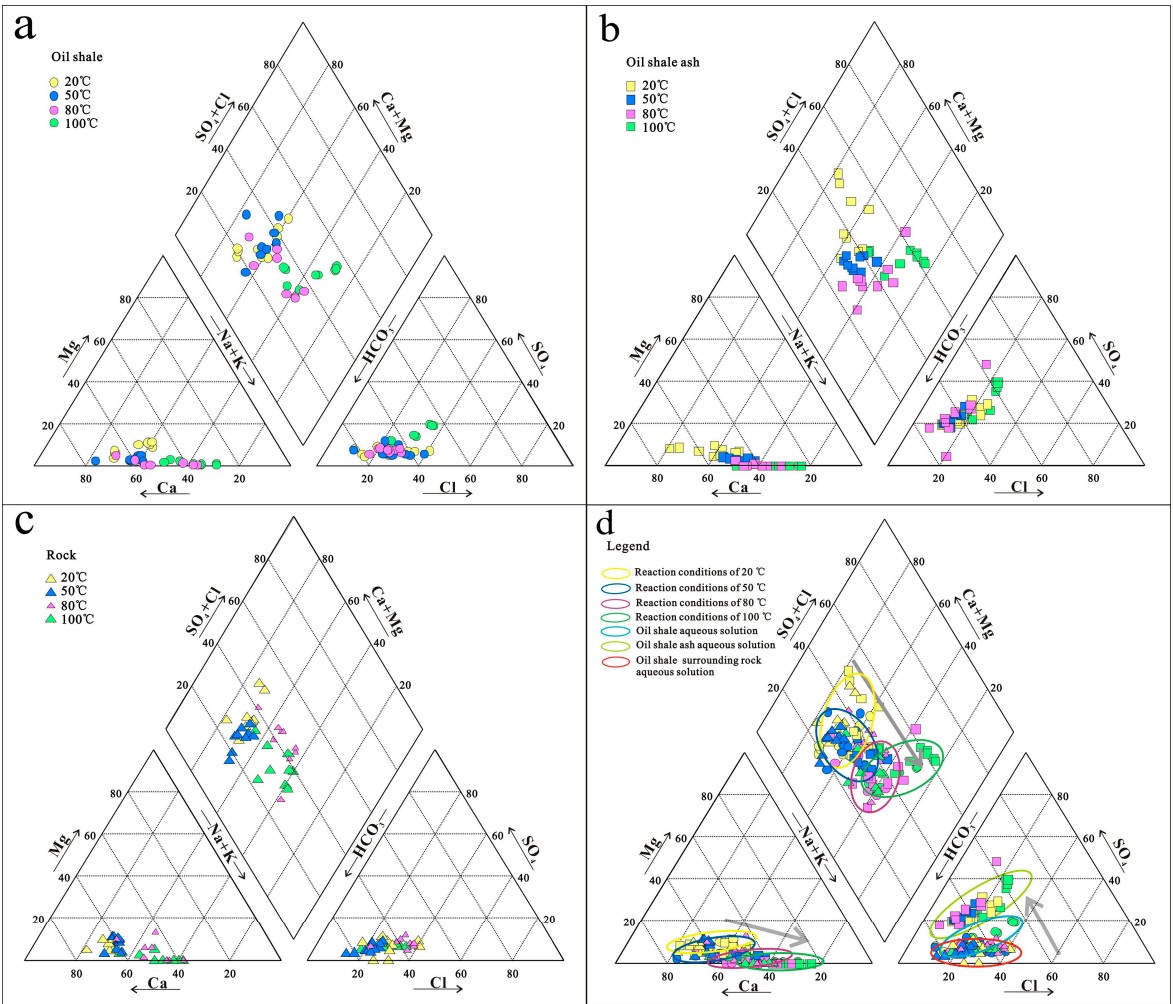

**Figure 10.** Piper ternary diagrams of water–rock interactions in the aqueous solution with different reaction conditions. (**a**–**c**) show the chemical compositions of water–rock interactions in oil shale aqueous solution, oil shale ash aqueous solution and rock aqueous solution, respectively. (**d**) shows the chemical compositions of these there aqueous solutions together.

*4.3. Formatting*

The variance contribution rate of the first principal Factor F1 was 33.903%, which was mainly composed of $K^+$, $Na^+$, $HCO_3^-$, and $SO_4^{2-}$. There was a strong correlation among these four ions, indicating that they may have the same material source or formation process. According to the XRD, $K^+$ in the aqueous solution mainly came from the dissolution of illite $(KAl_2[(SiAl)_4O_{10}]\cdot(OH)_2\cdot nH_2O)$:

$$K_{0.5}Mg_{0.25}Al_{2.3}Si_{3.5}O_{10}(OH)_2 + 11/10H^+ + 63/20H_2O \rightarrow 23/20Al_2Si_2O_5 + 1/2K + 1/4Mg^{2+} + 6/5Si(OH)_4 \quad (1)$$

$Na^+$ came from the dissolution of plagioclase $(Na[AlSi_3O_8]\text{-}Ca[Al_2Si_2O_8])$ and montmorillonite $((Na,Ca)_{0.33}(Al,Mg)_2[Si_4O_{10}](OH)_2\cdot nH_2O)$:

$$3Na_{1/3}Al_{7/3}Si_{11/3}O_{10} + 30H_2O + 6OH^- \rightarrow Na^+ + 7Al(OH)_4^- + 10H_4SiO_4 \quad (2)$$

$$NaAlSi_3O_8 + H^+ + 9/2H_2O \rightarrow 1/2Al_2Si_2O_5(OH)_4 + Na^+ + Si(OH)_4 \quad (3)$$

$HCO_3^-$ came from the dissolution of dolomite $(CaMg(CO_3)_2)$ and calcite $(CaCO_3)$:

$$CaCO_3 + CO_2 + H_2O \rightarrow Ca^{2+} + 2HCO_3^- \quad (4)$$

$$CaMg(CO_3)_2 + CO_2 + H_2O \rightarrow Ca^{2+} + Mg^{2+} + 4HCO_3^- \quad (5)$$

$SO_4^{2-}$ came from the dissolution of gypsum $(CaSO_4\cdot 2H_2O)$ or other sulphates and the oxidation of pyrite $(FeS_2)$:

$$2FeS_2 + 7O_2 + 2H_2O \rightarrow 2FeSO_4 + 4H^+ + 2SO_4^{2-} \quad (6)$$

Therefore, F1 reflects the influence of the dissolution of minerals in the sample on the chemical composition of the groundwater.

The variance contribution rate of the second main Factor F2 was 17.71%, which was mainly composed of the temperature, $Mg^{2+}$, and $Ca^{2+}$ variables. According to the XRD analysis, $Ca^{2+}$ and $Mg^{2+}$ in the aqueous solution mainly came from the dissolution of dolomite $(CaMg(CO_3)_2)$ and calcite $(CaCO_3)$. Therefore, F2 indicates that temperature had a great influence on the dissolution of this kind of mineral. However, $HCO_3^-$ did not belong to the same factor as $Mg^{2+}$ and $Ca^{2+}$, indicating that $HCO_3^-$ may have other sources. The proportion coefficient can be used to analyze the origin and evolution characteristics of hydrochemical components, and the molar ratio of different ions can be used to describe the hydrochemical characteristics of different groundwater masses to further identify the lithology and origin of the components flowing through the aquifer. In general, if $Ca^{2+}$, $Mg^{2+}$, and $HCO_3^-$ in water are mostly derived from carbonate minerals, then the ratio of $(Ca^{2+} + Mg^{2+})$ to $HCO_3^-$ should theoretically be equal to 1 [29]. Figure 11 depicts the ion ratio relationship among $Ca^{2+}$, $Mg^{2+}$, and $HCO_3^-$. The results show that, in the water–rock interaction solution, at a reaction temperature of 20 °C, only some of the samples $(Ca^{2+} + Mg^{2+})/HCO_3^-$ ranged from 1:1 to 1:2, and most of the sample points were below 1:2, indicating that the dissolution of dolomite $(CaMg(CO_3)_2)$ and calcite $(CaCO_3)$ was only part of the source of $HCO_3^-$. Therefore, the solubility of $Na_2CO_3$ should be taken into consideration. When the concentration of $Na^+$ in an aqueous solution is high, the content of $HCO_3^-$ will exceed the upper limit as related to $Ca^{2+}$.

The variance contribution rate of the third principal Factor F3 was 12.941%, which mainly consisted of the reaction time, Cl, $F^-$, and $Fe^{2+}$ variables. In these aqueous solutions, $Cl^-$ mainly came from the dissolution of rock salt (NaCl) or other chlorides $(MgCl_2, CaCl_2)$. Figure 12 is the scatter plot showing the $\gamma Na/\gamma Cl$ ratios in the aqueous solution under different reaction conditions. The results show that the content of $Na^+$ in the solution was higher than that of $Cl^-$ (the ratio of $Na^+/Cl^-$ is mostly below the 1:1 line), indicating that rock salt was not the only source of sodium ions in the solution.

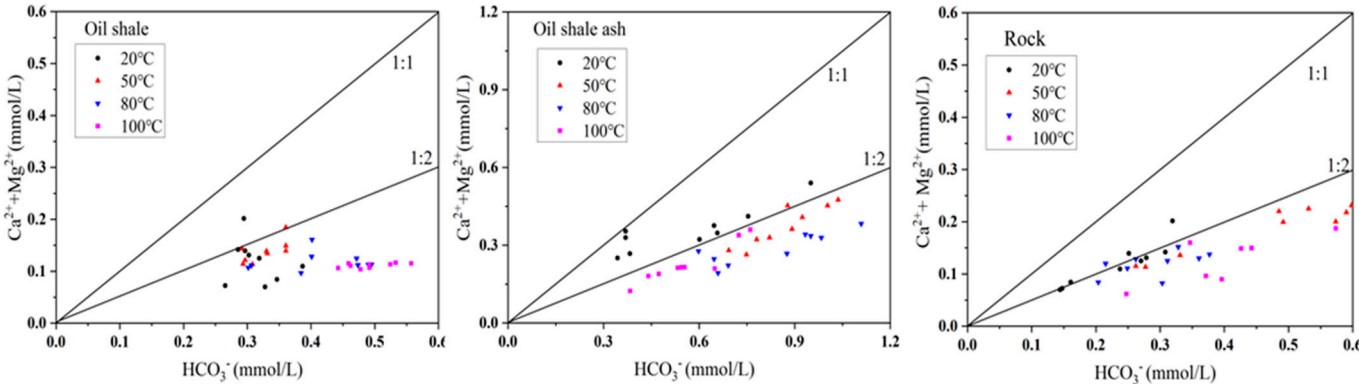

**Figure 11.** The ratio of $Ca^{2+}$, $Mg^{2+}$, and $HCO_3^-$ in the aqueous solution under different reaction conditions.

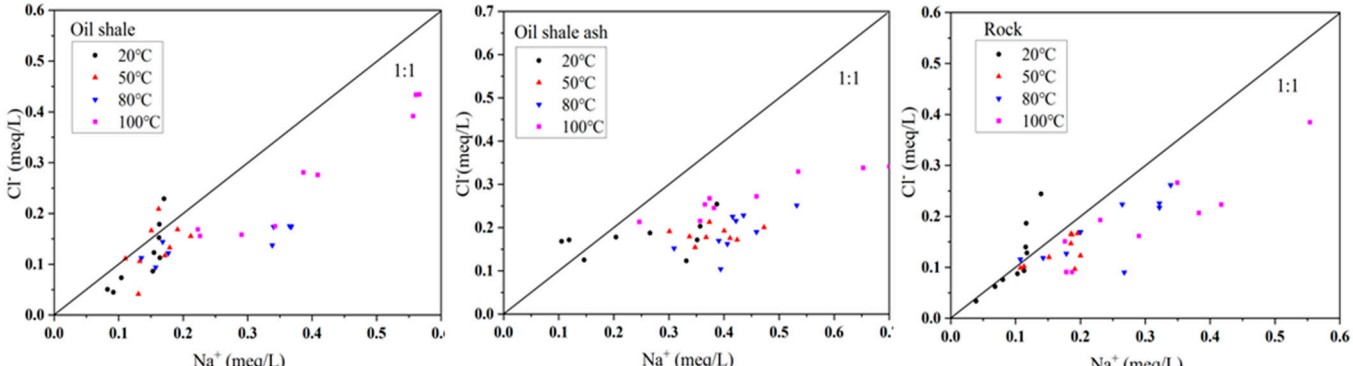

**Figure 12.** Scatter diagram of $\gamma Na/\gamma Cl$ in the aqueous solution under different reaction conditions.

According to the XRD results, there was a large amount of pyrite ($FeS_2$) in both the oil shale and the surrounding rock, which was the main source of iron in the aqueous solutions. $F^-$ was mainly derived from the hydrolysis of fluorinated minerals such as fluorite ($CaF_2$) and apatite ($Ca_5(Cl,F,OH)(PO_4)_3$). Therefore, F3 indicates that the length of the reaction time had a great influence on the dissolution of such minerals.

The variance contribution rate of the fourth main Factor F4 was 10.669%, which was mainly composed of the lithology and $NO_3^-$ variables. The content of $NO_3^-$ in the aqueous solution was very low and came mainly from the degradation of nitrogenous organic matter. Oil shale is a mineral rich in organic matter. The heating and retorting process of oil shale is also the process of extracting organic matter from oil shale, so it will cause the differences in organic matter content among the samples. Therefore, F4 indicates that the organic matter content in different rock types varies greatly.

## 5. Conclusions

A series of experiments on the water–rock interaction were conducted to analyze the influence of oil shale on the groundwater environment during in situ exploitation. Further, to identify the main factors affecting the variation of the groundwater ion concentration, the mineral composition in the oil shale and surrounding rock was tested, accompanied by a statistical analysis of the experiments' results. Based on the results of the work, the following conclusions were drawn:

(1) With the increase of the temperature and reaction time, the content of the ions in the three aqueous solutions increased. The $Ca^{2+}$ content in the aqueous solution was the highest under the 20 °C and 50 °C temperature conditions, and $Na^+$ became the principal cation at 80 °C and 100 °C. $HCO_3^-$ was always the main anion in the aqueous solution under the different reaction conditions.

(2) Under the different reaction conditions, the ion content in the oil shale–ash aqueous solution changed the most. The water quality of the oil shale–ash aqueous solution

was changed into HCO<sub>3</sub>-SO<sub>4</sub>-Na-K at 80 °C and 100 °C. This indicates that the pyrolytic oil shale was more likely to participate in the reaction when it came into contact with the water. Attention should be paid to the isolation of reservoirs and aquifers after oil shale in situ exploitation.

(3)  The source of the main components in the aqueous solution was identified by factor analysis. The results showed that the mineral type had the greatest influence on the ionic components, followed by the temperature, reaction time, and organic matter content in the rocks.

In short, oil shale in situ exploitation will affect the inorganic environment of adjacent aquifers. Because of the geological environments of oil shale areas are different, the compositions of oil shale and its surrounding rocks are also different, so the impacts of in situ mining on groundwater inorganic minerals should be taken into consideration when evaluating in situ exploitation projects for oil shale.

**Author Contributions:** Conceptualization, S.H.; writing and editing, Q.L.; methodology, L.L.; analysis, Q.L. and S.H.; supervision, Q.Z.; project administration, S.H. All authors have read and agreed to the published version of the manuscript.

**Funding:** This research was funded by the National Natural Science Foundation "Study on the mechanism of BTEX release and aquifer pollution during in situ exploitation of oil shale" of China (Grant No. 42002260).

**Institutional Review Board Statement:** Not applicable.

**Informed Consent Statement:** Not applicable.

**Data Availability Statement:** Not applicable.

**Acknowledgments:** The authors acknowledge the support of The National Natural Science Funds of China (42002260). And we are thankful to Editors, Reviewers and Editorial Board for their constructive comments, which improved the manuscript significantly.

**Conflicts of Interest:** The authors declare no conflict of interest.

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
