# Peer review of "Impact of Inorganic Solutes’ Release in Groundwater during Oil Shale In Situ Exploitation"

_water, doi:10.3390/w15010172_

Round 1

Reviewer 1 Report

Impact of Inorganic Mineral Release on Groundwater System During Oil Shale In-Situ Exploitation

This MS has the fair potential to publish in the Energies after revision. It focuses on the Impact of Inorganic Mineral Release on Groundwater System During Oil Shale In-Situ Exploitation.  What is novelty, it looks like just report. 

Abstract 

The results show that the release of inorganic minerals from oil shale to groundwater due to the heat released during the in-situ exploitation of oil shale cannot be ignored.

On what basis you decided about heat released, it is suggested to rewrite this section in professional scientific way to make it more readable for readers.

Revise your abstract.

*       What are the key results?

*       What are the practical implications of your research (how can the results be utilized by e.g., readers, community, or companies)?

Introduction

Your article must answer the following basic questions:

*       The problem statement under investigation, methodology, and results/and discussion must be aligned.

*       What are your results?

*       What are the implications of the results?

*       What do you recommend as a further study for others?

Available Dataset

Narrate all the data set you used in a separate section "Available data set". Remove the dataset from the methodology part. Briefly discuss the data and methodology.

2.1. Study Area and Samples

What is source of information to build this section? it’s your own work or take from literature?? If you take some material form literature, it is required to properly cited.

2.2. Experiment of Water-rock Interaction

On what basis you adopt this experiment, it is requested to properly provide its scientific justification.

2. Materials and Methods

 This section is full of builders, it is suggested to rewrite this section in proper way.

Figure 1. Conceptual model for oil shale in-situ exploitation.

What is source of data of this figure? it’s your own development or take from literature??

Author Response

Dear Professor,

Thank you for your comments and suggestions. I have finished the modification according to your suggestion. And the point-by-point responses are in the attachment.

  1. Abstract 

The results show that the release of inorganic minerals from oil shale to groundwater due to the heat released during the in-situ exploitation of oil shale cannot be ignored.

On what basis you decided about heat released, it is suggested to rewrite this section in professional scientific way to make it more readable for readers.

Revise your abstract.

*       What are the key results? 

*       What are the practical implications of your research (how can the results be utilized by e.g., readers, community, or companies)?

Reply: I've rewritten the Abstract.

Thanks for your review. According to your comments, in the abstract, I introduced the background of the research, and then the research purpose, research methods and key results were wrote orderly. At last, the importance of the research was expounded and some suggestions on environmental protection of in-situ exploitation of oil shale were put forward.

I need to explain that oil shale is usually pyrolyzed at 300℃ to 500℃ during in-situ exploitation, but the formation is still insulated after the shale oil is extracted. This is a very wide temperature range. In our previous studies we found that not only oil shale in situ mining process make groundwater organic pollution aggravate but also continuous pollution of groundwater caused by the residual oil shale ash still exist (The influence of oil shale in situ mining on groundwater environment: A water-rock interaction study[J]. Chemosphere, 2019, 228(AUG.):384-389.). In this study, we want to explore the inorganic solutes transport process during thermal oil shale in-situ exploitation.

  1. Introduction

Your article must answer the following basic questions:

*       The problem statement under investigation, methodology, and results/and discussion must be aligned.

*       What are your results?

*       What are the implications of the results?

*       What do you recommend as a further study for others?

Reply: I have rewritten the introduction according to the revision suggestions

Thank you for your valuable advice. The introduction section was a bit confusing because it tried to express too much.

Therefore, according to your suggestions, I have revised the introduction: In the first paragraph, I introduce oil shale and in-situ oil shale exploitation techniques, and suggest that research on the impact of in-situ oil shale exploitation on the underground environment needs to be considered.

In the second paragraph, I describe the hydraulic connection between oil shale reservoirs and aquifers that can occur from in-situ exploitation of oil shale

In the third paragraph, I describe the current researches on the impact of in-situ oil shale production on groundwater and point out the shortcomings.

In the fourth paragraph, based on the above discussion, I introduce my research plan and data analysis method, and emphasize the research purpose again.

With the revision of the introduction, The problem statement under investigation,methodology, and results/and discussion are already consistent.

  1. Available Dataset

Narrate all the data set you used in a separate section "Available data set". Remove the dataset from the methodology part. Briefly discuss the data and methodology.

Reply: According to your suggestion, I added “2.4 Analysis method of data” to “2. Materials and Methods”, which mainly describes the data analysis method adopted in the article and the feasibility of this method used to analyze my experimental data.Then the analysis results are discussed in the discussion section.

  1. 1. Study Area and Samples

What is source of information to build this section? it’s your own work or take from literature?? If you take some material form literature, it is required to properly cited.

Reply: Thank you for reminding me. The information for this part is obtained from other articles and has been noted and quoted in the article.

  1. 2. Experiment of Water-rock Interaction

On what basis you adopt this experiment, it is requested to properly provide its scientific justification.

Reply: The in-situ processing of oil shale requires a long heating time; for example, Shell’s in-situ conversion process (ICP) utilizes electricity to heat oil shale underground over the period of two years. Although the whole oil shale utilisation process adversely affects the environment, groundwater included, yet to date, the effect of its in-situ conversion processing on the groundwater has not been fully clarified. Water-rock interaction means that the groundwater reacts with ambient rock formations mechanically, chemically or physically, so that affecting and altering the chemical composition and properties of water as well as the states of rock formations. This study mainly focuses on the interaction between groundwater and oil shale. The water-rock interaction experiment can be used to study the release of Inorganic matter from groundwater during oil shale In-situ exploitation.

  1. Materials and Methods

This section is full of builders, it is suggested to rewrite this section in proper way.

Reply: Thank you for your suggestion. I have modified this part, which mainly introduces the research area, experimental scheme, sample test method and data processing.

  1. Figure 1. Conceptual model for oil shale in-situ exploitation.

What is source of data of this figure? it’s your own development or take from literature??

Reply: Thanks for your advice.The information for this part is obtained from other articles and has been cited in the article.

Reviewer 2 Report

Review of “Impact of inorganic mineral release on groundwater system during oil shale in-situ exploitation” by Li et al.

Major comments: This paper contains some interesting applied science to the management of aquifer water quality in systems occurring adjacent to thermal extraction of petroleum from oil shales. It merits publication, but will require major changes to meet the standard of the journal for publication.

The title should be changed slightly to better describe the research. I suggest “Impact of inorganic solutes released in groundwater during thermal oil shale in-situ exploitation”. The abstract really does not capture the important results of the research and should be rewritten. A key aspect is to answer the question, So What!. What was the research problem, why is it important, and what have you found that advances understanding.

There are numerous required changes to the English in the text and I have made a PDF of suggested changes to help guide your revision. The changes are hand written and in red color.

A major issue is the quality of figures 4 to 13. They are not readable at the scales presented. Either increase the sizes of the figures or redraft them with larges fonts and graph points so they can be clear.

Within the Methods sections, you needs to add a description of the statistical methods applied to analysis of the data. How you applied the factor analysis is unclear to the reader.

Within the Discussion, section 4.1 needs to be rewritten. The sentence structure is awkward and it is mixed. Perhaps you should have a native English speaker read and help revise this section. There are too many sentences linked together, which makes it very different to read.

In section 4.2, there are some statistical test data presented. However, there is no clear explanation of the importance or significance of the values and how they relate to the issue of groundwater contamination. Please revise this section.

In Figures 12 and 13, a series of scatter plots are presented. In several of these plots, a best fit line is shown. What is the R2 and p-values for these lines and are they statistically significant?

Author Response

Dear professor,

I have finished the modification according to your suggestion. Thanks for your comments and suggestions, they are helpful. And the point-by-point responses are  as follow.

  1. The title should be changed slightly to better describe the research. I suggest “Impact of inorganic solutes released in groundwater during thermal oil shale in-situ exploitation”.

Reply: Thanks for your suggestion. I have changed the title to “Impact of inorganic solutes released in groundwater during thermal oil shale in-situ exploitation”.

  1. The abstract really does not capture the important results of the research and should be rewritten. A key aspect is to answer the question, So What!. What was the research problem, why is it important, and what have you found that advances understanding.

Reply:Thanks for your suggestion. You are right, the abstract does not highlight the main point of the article. According to your and the first reviewer’s comments, I have rewritten the abstract as follow:

Abstract: Oil shale can produce oil and shale gas by heating oil shale at 300℃-500℃. The high temperature and the release of organic matter can change the physical and mechanical properties of rocks, and make the originally tight impervious layer become a permeable layer under in-situ exploitation conditions. To realize the potential impact of in-situ exploitation of oil shale on groundwater environments, a series of water-rock interaction experiments under different temperatures is conducted. The results show that with the increase of reaction temperature, the anions and cations in the aqueous solution of oil shale, oil shale ash, and the surrounding rock show different trends, and the release of anions and cations in the oil shale ash solution is most affected by the ambient temperature. The hydrochemical type of oil shale ash solution is SO4·HCO3-Na·K at 80℃ and 100℃, which changes into the sulfuric acid type and changes the water quality. The main reasons are 1), The high temperature (≥80℃) can promote the dissolution of FeS in oil shale; 2), The porosity of oil shale increases after pyrolysis and make it easier to react with water. This paper is an important supplement to the research on the impact of in-situ exploitation of oil shale on the groundwater environment. Therefore, the impacts of in-situ mining on groundwater inorganic minerals should be taken into consideration when evaluating in-situ exploitation projects of oil shale.

  1. There are numerous required changes to the English in the text and I have made a PDF of suggested changes to help guide your revision. The changes are hand written and in red color.

Reply: Thank you for your suggestion. I have corrected the English errors in the article according to your PDF.

  1. A major issue is the quality of figures 4 to 13. They are not readable at the scales presented. Either increase the sizes of the figures or redraft them with larges fonts and graph points so they can be clear.

Reply: According to your comments, I have made the lines of the picture bold and improved the quality of the picture. But the colors used in hydrogeological maps were drawn according to national standards, and certain color represent certain strata, so I think it's not appropriate to change the colors. But I've bolded the boundaries of different strata. Hopefully it will be well read in the modified version.

  1. Within the Methods sections, you needs to add a description of the statistical methods applied to analysis of the data. How you applied the factor analysis is unclear to the reader.

Reply: Thank you for your comments. I have added a description of the statistical methods applied to data analysis to the methods section (2.4) and explained the meaning of the indicators.

  1. Within the Discussion, section 4.1 needs to be rewritten. The sentence structure is awkward and it is mixed. Perhaps you should have a native English speaker read and help revise this section. There are too many sentences linked together, which makes it very different to read.

Reply: We are very sorry for our incorrect writing, the writing problem in the manuscript has been revised.

  1. In section 4.2, there are some statistical test data presented. However, there is no clear explanation of the importance or significance of the values and how they relate to the issue of groundwater contamination. Please revise this section.

Reply: According to your comments, I have added some explanations of the meaning of values in section 4.2, and expounded their relationship with groundwater problems:

Table 2 show the eigenvalues and cumulative variance contribution rates of the factor correlation matrix calculated by factor analysis. It can be seen from the factor contribution rate that the eigenvalues of the first four factors were greater than 1, and the sum of the eigenvalues of the first four factors accounted for 75.223% of the total eigenvalues, which means 75.223% information of the total sample can be reflected by the four factors. So these four factors awere extracted as the main factors.

Table 3 show the factor loading after rotation. Factor load is the correlation coefficient between a variable and the factor. For a variable, the larger the absolute value of the load is, the closer the relationship between it and the factor is.

  1. In Figures 12 and 13, a series of scatter plots are presented. In several of these plots, a best fit line is shown. What is the R2 and p-values for these lines and are they statistically significant?

Reply: Thank you for your comments. But the curves presented in figures 12 and 13 are not fitting curve, they are only ratio line of 1:1 and 1:2, which is used to observe and judge whether the canion and anion meet the distribution of 1:1 or 1:2. So R2 and P-values don't exist actually.

Round 2

Reviewer 1 Report

I do not think it has the potential to publish in water. This paper has very limited information. 

Author Response

Dear Professor,

I'm sorry to hear that. But I think this manuscript has the potential to publish in water. Because our previous study (organic contamination of in-situ oil shale exploitation on aquifer system) has been accepted by water. This study is a extension  and complement of previous research. But thank you for your valuable opinions of round 1. They improved the quality of the manuscript in a great extent. Thanks for your work.

Reviewer 2 Report

I suggest carefully editing the revised text to eliminate any errors. The figures have been improved, but check the font sizes in the last two figures. They could be increased in size to make the figure more readable. The references are not in the journal style and should be corrected.

Author Response

Dear Professor,

Thanks for your help. I've bolded and adjusted the fonts on the last two figures to make sure they're easy to read. And the references have been re-edited according to water's requirements. I went over the manuscript with my tutor and corrected some grammatical errors. Thank you again for your comments.

Yours Sincerely,

Qingyu Li
